# Synthesis of pH and Glucose Responsive Silk Fibroin Hydrogels

**DOI:** 10.3390/ijms22137107

**Published:** 2021-07-01

**Authors:** Xiaosheng Tao, Fujian Jiang, Kang Cheng, Zhenzhen Qi, Vamsi K. Yadavalli, Shenzhou Lu

**Affiliations:** 1National Engineering Laboratory for Modern Silk, College of Textile and Clothing Engineering, Soochow University, Suzhou 215123, China; 20194215005@stu.suda.edu.cn (X.T.); 20175215006@stu.suda.edu.cn (F.J.); 20185215008@stu.suda.edu.cn (K.C.); 20204215016@stu.suda.edu.cn (Z.Q.); 2Department of Chemical & Life Science Engineering, Virginia Commonwealth University, Richmond, VA 23284, USA; vyadavalli@vcu.edu

**Keywords:** silk fibroin, hydrogel, ε-Poly-(L-lysine), pH-responsive, glucose-responsive

## Abstract

Silk fibroin (SF) has attracted much attention due to its high, tunable mechanical strength and excellent biocompatibility. Imparting the ability to respond to external stimuli can further enhance its scope of application. In order to imbue stimuli-responsive behavior in silk fibroin, we propose a new conjugated material, namely cationic SF (CSF) obtained by chemical modification of silk fibroin with ε-Poly-(L-lysine) (ε-PLL). This pH-responsive CSF hydrogel was prepared by enzymatic crosslinking using horseradish peroxidase and H_2_O_2_. Zeta potential measurements and SDS-PAGE gel electrophoresis show successful synthesis, with an increase in isoelectric point from 4.1 to 8.6. Fourier transform infrared (FTIR) and X-ray diffraction (XRD) results show that the modification does not affect the crystalline structure of SF. Most importantly, the synthesized CSF hydrogel has an excellent pH response. At 10 wt.% ε-PLL, a significant change in swelling with pH is observed. We further demonstrate that the hydrogel can be glucose-responsive by the addition of glucose oxidase (GOx). At high glucose concentration (400 mg/dL), the swelling of CSF/GOx hydrogel is as high as 345 ± 16%, while swelling in 200 mg/dL, 100 mg/dL and 0 mg/dL glucose solutions is 237 ± 12%, 163 ± 12% and 98 ± 15%, respectively. This shows the responsive swelling of CSF/GOx hydrogels to glucose, thus providing sufficient conditions for rapid drug release. Together with the versatility and biological properties of fibroin, such stimuli-responsive silk hydrogels have great potential in intelligent drug delivery, as soft matter substrates for enzymatic reactions and in other biomedical applications.

## 1. Introduction

Diabetes is a non-communicable, chronic metabolic disease often leading to serious long-term complications such as retinopathy, renal failure, neuropathy and cardiovascular disease. Worldwide, there are over 460 million adults suffering from diabetes. This number is expected to exceed 700 million by 2050 [1]. During the current global pandemic, it has been observed that high blood glucose concentrations in diabetic patients may not only make them more susceptible to COVID-19 infection, but also enhance the replication and expression of the virus. This has led to more serious clinical symptoms among diabetic patients with COVID-19 infection [2,3,4,5]. The control and management of blood glucose levels in diabetic patients is, therefore, of great importance. This process involves real-time monitoring of blood glucose levels, and, often, multiple injections of insulin to ensure effective treatment [6,7,8]. However, frequent monitoring and administration is often painful for patients, resulting in poor blood glucose management. Owing to the complexity of the physiological environment, automated delivery systems may lead to improper blood glucose control, or life-threatening hypoglycemia due to incorrect calculations of the insulin dose [6]. Chemically-controlled closed-loop insulin delivery has been suggested as a treatment alternative. This can include delivery of insulin via a glucose-responsive material, or by modifying insulin with glucose-sensitive functional groups to trigger active release [9,10,11].

Glucose-responsive hydrogels in particular, have received attention owing to their chemical versatility and stimuli-responsive behavior [12]. Hydrogels are polymers with a three-dimensional network and interconnected porous structure, high water retention and swelling properties. When external or internal stimuli (such as light, temperature, electric field and pH) are sensed, such materials react and change their structure, morphology, or physicochemical properties, which may be correlated to the corresponding triggering function [13,14,15]. Compared to traditional whole body administration, the local administration of hydrogels has advantages including: (i) stabilization of drug molecules and maintenance of activity, (ii) control and extended release of drug molecules and (iii) direct delivery to the disease site, reducing waste and toxic side effects [14]. Glucose-responsive hydrogels using glucose binding protein (GBP) [16,17,18], phenylboronic acid [19,20] or glucose oxidase (GOx) [21] have been proposed for the regulation of blood glucose. Hydrogels based on GOx are of particular interest owing to their convenient use and excellent safety.

In comparison to synthetic hydrogels, bio-based stimuli-responsive hydrogels are attractive owing to their biocompatibility, inherent biological activity and enzymatic degradation [13,22]. Silk fibroin (SF), as a versatile natural structural protein [23], has been widely used in tissue engineering scaffolds [24,25], wound dressings [26,27] and drug delivery [28,29,30,31] because of its excellent mechanical properties, biocompatibility, controllable biodegradability and low immunogenicity [32,33,34]. However, due to the presence of acidic aspartic acid and glutamic acid residues, SF chains are negatively charged, limiting their pH response [35]. The pH within the human body, e.g., gastrointestinal tract, vagina, blood vessels etc., changes significantly, making the development and design of an SF based biomaterial with pH response very useful, albeit challenging. pH-responsive materials have the ability to swell in solutions of acidic pH and de-swell or shrink in solutions of alkaline pH. This reversible swelling-shrinking property brought about by changes in external pH makes these materials useful in a wide range of applications such as drug delivery systems and chemical sensors [36]. Application of cationic silk fibroin in skin scaffolds and gene delivery has been demonstrated [37,38], but its potential pH-stimuli responsiveness has not been elaborated. At the same time, there are no reports of silk fibroin gel as a soft substrate for enzyme reaction in a cationic hydrogel microenvironment.

Here, we use ε-Poly-(L-lysine) (ε-PLL), a cationic polypeptide with excellent antibacterial properties, to impart pH-responsive behavior in bio-based SF hydrogels [39,40,41,42]. ε-PLL is a product of the combination of a carbonyl group at the α-position and an amino group at the ε-position of lysine, an essential amino acid [43,44,45]. ε-PLL has been previously used as a biodegradable glucose-responsive cationic polymer for constructing polymer–insulin complexes for glucose-stimulated insulin delivery [46]. In this study, SF was modified with ε-PLL to form pH-responsive cationic hydrogels (CSF) by enzymatic crosslinking reaction and its response to different pH values was demonstrated. Morphological and physico-chemical characterization of the hydrogels was performed. We further explored the response of the CSF hydrogels to glucose concentration via incorporation of the enzyme glucose oxidase. The rapid diffusion mechanism driven by high swelling provides the possibility for the intelligent transport of drugs, and the possibility for closed-loop detection and treatment of diabetes in vivo [47]. As shown in Figure 1, we have designed a new type of pH and glucose responsive silk fibroin hydrogels.

## 2. Results

### 2.1. Cationic Modification of Silk Fibroin from Bombyx Mori

The isoelectric point of SF is ~4.1, because the glutamic acid and aspartic acid side chain -COOH groups of SF lose H^+^ in neutral solution and form -COO^−1^, making it negatively charged. Therefore, our goal is to obtain positively-charged SF molecules sensitive to acidic pH by grafting ε-PLL (a cationic polypeptide composed of 25–30 lysine residues). Figure 2 shows that carbodiimide (EDC) and N-hydroxysuccinimide (NHS) are used to activate the -COOH group on the side chain of SF. The amino group on positively-charged ε-PLL is then grafted onto the side chain of SF to obtain the CSF.

Figure 3a shows the change of potential of modified SF solution with the change of mass of ε-PLL. The ε-PLL accounted for 0%, 0.5%, 1%, 2%, 5%, 10% and 15 wt.% of SF, respectively, and reacted with 50 mL of 30 mg/mL SF solution. It can be noted that the zeta potential of SF modified by ε-PLL shows an upward trend with the increase of ε-PLL content from 0% to 15 wt.%. After the addition of ε-PLL, the zeta potential of SF changes from a negative value (−7 mV) to positive, and with the increase of ε-PLL content, the zeta potential gradually increases. This indicates that the -COOH group on the side chain of SF participates in the reaction, resulting in a decrease of acidic groups and an increase of basic groups, thereby increasing the positive charge. When the mass of the added ε-PLL accounts for 10 wt.% and 15 wt.% of the SF mass, the zeta potential becomes ~7 mV and no longer increases. This is likely because the carboxyl groups in SF are limited, and the reaction becomes saturated. Further experiments showed that when 20 wt.% ε-PLL was added, the SF solution gelatinized and could not react further. Therefore, in subsequent experiments, 10 wt.% ε-PLL was used to graft SF. The zeta potential of the modified SF is ~6.5 mV.

The isoelectric point (pI) of pure SF solution is ~4.1 (Figure 3b), while the pI of SF solution grafted with 10 wt.% ε-PLL increases significantly to ~8.6 (Figure 3c). This increase indicates that ε-PLL is successfully grafted onto SF, consistent with the previous zeta potential results. Figure 3d shows the gel electrophoresis of SF before and after cationic polymerization. Lane 2 is the molecular weight distribution band of pure SF; lane 3 is the molecular weight distribution band of pure ε-PLL. The results show that the molecular weight distribution of lane 4 is the same as that of lane 2, which indicates that modified SF cannot be achieved by simply blending SF solution with ε-PLL solution. The molecular weight distribution of lane 5 can be seen by ε-PLL solution. The molecular weight of the grafted SF was 50~100 kDa.

In order to study the effect of grafting on the aggregation of SF, the infrared absorption spectrum and XRD before and after grafting were characterized. As seen in Figure 4a, the characteristic peaks of pure SF at 2979 cm^−1^, 3065 cm^−1^ belong to the -COOH group (3300–2500 cm^−1^) [36,48]. After the SF was grafted with different content of ε-PLL, the characteristic peak of -COOH group (3065 cm^−1^) gradually decreased. The characteristic peak of amino group (3500–3300 cm^−1^) [49] gradually becomes sharper with an increase in mass fraction of ε-PLL. This can be attributed to the fact that more amino groups are successfully introduced [36,48,49] and that ε-PLL was successfully grafted onto SF to achieve the CSF. However, in the range of amide I (1600–1700 cm^−1^), the absorption peaks of pure SF and different ε-PLL grafted groups appear around 1642 cm^−1^, which are mainly caused by stretching vibration. In the range of 1200–1300 cm^−1^, the absorption peak appears at 1239 cm^−1^, which is mainly C-N stretching vibration and N-H bending vibration. From the IR absorption spectrum of the amide interval, it is noted that the grafting of ε-PLL does not change the aggregation structure of the SF solution, and the CSF in the form of the solution still maintains a random coil structure [49,50]. Our results are in excellent agreement with this phenomenon which was previously reported by Sahoo et al. [51].

It can be seen from Figure 4b that pure SF is mainly a random coil structure, with a large wide diffraction peak at 20.7° and a small diffraction peak at 9.1°, indicating a small amount of Silk II crystal structure. Compared with the pure SF diffraction curve, the diffraction peak remains unchanged with the increase of ε-PLL percentage, and a larger diffraction peak is formed at 20.7°, indicating that CSF is mainly a kind of irregular coil structure [52,53]. The results show that ε-PLL modification does not have a significant effect on the overall crystal structure of SF.

### 2.2. Morphology of CSF Hydrogel

Figure 5 shows the microscale morphology of the different hydrogel samples. All the hydrogels can maintain their shape and are soft and elastic. CSF hydrogels modified with ε-PLL are transparent, while hydrogels formed by spontaneously blending ε-PLL and pure SF hydrogel are milky white. After adding GOx, the hydrogels are yellow and transparent, whereas the CSF hydrogels are transparent. Electron microscopy images of pure SF and CSF hydrogels are shown in Figure 5a–f. The porous structure of CSF hydrogel is observed, and the cross-sectional morphology changes with different proportions of ε-PLL grafted. The cross-sectional morphologies also show that the intrinsic structure of CSF hydrogels is different from that of traditional SF hydrogels [54,55]. Generally, CSF hydrogels exhibit more intertwined structural characteristics, which are shallow and deep, with different thickness of pore walls, dense arrangement and a cross-linked network structure. This kind of intertwined pore structure provides a suitable environment for the survival of living cells and the transport of beneficial ingredients [56]. At the same time, it has characteristics of absorbing large amounts of water, which reflects the significance of these reticular structures vis-a-vis swelling behavior.

### 2.3. Aggregation Structure of CSF Hydrogel

In order to study the structure change of CSF hydrogels, the infrared absorption spectra of these materials were studied. For example, Figure 6a shows that pure SF hydrogels have distinct absorption peaks at 1635 cm^−1^, 1528 cm^−1^ and 1250 cm^−1^ corresponding to the absorption peaks of amide I, amide II and amide III, which are characteristic peaks of β-sheet structure [52,55]. The absorption peaks of CSF hydrogel at the range of 1655 cm^−1^, 1545 cm^−1^, 1235 cm^−1^ were obvious as the absorption peaks of amide I, amide II and III, caused by C=O tensile vibration, N-H bending vibration and C-N tensile vibration. The results show that the hydrogel prepared in this paper is primarily random coil structure [25].

From the wide-angle X ray diffraction curve Figure 6b, the pure SF hydrogel has a very prominent diffraction peak at 20.7°, and there are sharp diffraction peaks at 24.3°. These peaks characterize the crystalline structure of Silk II [53,57]. The comparison between the two indicates that the CSF hydrogel has typical amorphous structure, in agreement with the results of the IR spectrum.

### 2.4. Mechanical Properties of CSF Hydrogels

Figure 7a shows that the compressive strength of CSF hydrogel formed by ε-PLL grafting increases with an increase of ε-PLL content. The compressive strength of the CSF hydrogel reached 54.75 ± 0.76 kPa, while the compressive strength of the pure silk hydrogel is 10.94 ± 1.73 kPa. Initiators initiate free radical polymerization, resulting in chemical crosslinking between SF molecules. This results in dense network entanglements, giving CSF hydrogels a higher compressive strength. Pure SF hydrogel contains many lamellar structures formed by β-sheets, making the hydrogels more brittle and easier to compress and break. The compression strength of CSF hydrogels with different ε-PLL ratios is different, as is the number of crosslinking points in each hydrogel. This results in varying degrees of cross-linking entanglement, so that their compressive strength is different. For instance, for the 10 wt.% ε-PLL modified CSF hydrogel material, the morphology reveals the most compact pores and the highest degree of molecular entanglement, conferring the highest compressive strength.

The compression spring-back of CSF hydrogels formed by grafted ε-PLL is better than 35% (Figure 7b), which is far better than that of a pure SF hydrogel (13.5 ± 3.2%). This is again because the β-sheet content results in a brittle texture, and, therefore, poor compression re-elasticity. However, different proportions of CSF hydrogels have more chemical crosslinks and higher crosslinking density. The network structure imparts excellent compression resilience, while the 10 wt.% CSF hydrogels have the best compression resilience of 65.2 ± 1.5%. The compression ratio of 15 wt.% CSF hydrogel decreases, because the content of SF in hydrogels decreases, affecting the compression resilience. This also indicates that this value is higher at a higher content of ε-PLL.

### 2.5. pH Sensitive Swelling Properties of CSF Hydrogels

One of the goals this work is to modulate the pH responsive characteristics of SF. As shown in Figure 8a, the swelling of pure SF hydrogel is very low. Moreover, there is an initial decline (degree of swelling is the smallest at pH = 6) followed by a rise with increase of pH (degree of swelling is the largest at pH = 9). In contrast, the CSF hydrogel with different ε-PLL grafting ratio shows a sensitivity to acidic pH, with a high degree of swelling at low pH. The swelling of CSF hydrogel with the highest content of ε-PLL (10 wt.%) was the highest, and the swelling degree reached ca. 345% at pH = 4. This is because of the high amine content of ε-PLL. After ε-PLL is grafted onto SF, there is an increase in H^+^ at low pH. The H^+^ enters the hydrogel to bind the amine group of ε-PLL, which increases the positive charge of CSF molecules and repels the like charge. At the same time, because the gel contains -OH, -COOH, -NH_2_ and other groups, the weakly ionized group -COOH in the SF undergoes proton transfer under different pH. In acidic media, -OH and -NH_2_ groups are easily protonated to form -H_3_O^+^ and -NH_3_^+^, which form the interaction between ions and molecules with water, strengthening the hydrophilic properties of the hydrogel. The lower the pH, the more H^+^, the greater the repulsion between like charges and the greater the swelling degree. With the increase of solution pH, the protonation of -OH and -NH_2_ groups decreases, the repulsion of the like charges decreases and the swelling degree decreases. Figure 8b shows the hydrogel formed by simple blending of SF and ε-PLL. As a control group, the swelling degree does not change with pH value. The mixed water gels also do not show pH sensitive swelling. These results show that the pH sensitive swelling cannot be achieved by simple blending, but only via covalent crosslinking.

### 2.6. Properties of Glucose-Responsive CSF Hydrogels

In order to induce a glucose-dependent response in the pH sensitive hydrogels, the enzyme glucose oxidase (GOx) was incorporated. This catalyzes the conversion of glucose into gluconic acid, converting glucose concentration into a pH signal to form an enzyme-mediated response. The hydrogel loaded with enzyme was tested in PBS at different glucose concentrations. The glucose concentrations tested were high (400 mg/dL), critical (200 mg/dL) and normal levels (100 mg/dL). Different glucose oxidase contents (0, 2, 4 mg/mL) in the hydrogels were studied. From Figure 9a, it is noted that the hydrogel without GOx (i.e., 0 mg/mL) does not show responsiveness to glucose, and pH value remains unchanged. This suggests that the addition of GOx is necessary for glucose response. From Figure 9b,c, it is noted that hydrogels covered by high glucose solution show significant glucose-responsiveness. In addition, the pH of high glucose solution decreases steadily and is stable ~4.5. Due to the enzymatic conversion of glucose to gluconic acid, the rate of pH decrease increases with an increase of glucose concentration. On the contrary, the pH value in low sugar or pure saline solution almost does not decrease. The higher the content of enzyme in the hydrogel, the stronger the responsiveness of glucose and the faster the decrease in pH.

To explore the response to glucose and the change of pH in the internal and external solutions of xerogel and the change of pH values between internal and external solutions of xerogels, pH values and swelling of the xerogels were investigated with different glucose concentrations inside and outside the xerogels. As seen in Figure 9d, the response of xerogel to different glucose levels is clear. The pH of the external solution remains unchanged by the decrease of internal pH. The results further show that no H^+^ diffuses into the gel interior due to the higher pH value of the whole solution, reducing the pH response. This may be due to the continuous reaction of internal GOx, which supplies H^+^ ions. Even if some H^+^ ions diffuse into the external solution, the diffusion has a certain rate, which can promote an equilibrium pH value.

The results further verify that a CSF hydrogel with added GOx exhibits a good glucose-responsive swelling effect. CSF/GOx hydrogels show high swelling in high glucose environment and low swelling under low glucose concentration. There is no corresponding change in either SF/ε-PLL/GOx hydrogel or SF/GOx hydrogels in the control groups (Figure 9e). At a high glucose concentration (400 mg/dL), the swelling rate of CSF/GOx hydrogel was as high as 345 ± 16%, while swelling in 200 mg/dL, 100 mg/dL and 0 mg/dL glucose solutions was 237 ± 12%, 163 ± 13% and 98 ± 15% respectively. Comparatively, the swelling of the control group (SF/GOx and SF/ε-PLL/GOx hydrogels), under different glucose concentrations was not significant. Therefore, the aim of preparing glucose-responsive hydrogels is achieved.

## 3. Discussion

Multifunctional hydrogels capable of responding to both pH and glucose provide opportunities to form closed-loop delivery systems. Here the CSF hydrogels can respond to pH and via incorporation of the enzyme GOx, respond to glucose as well. GOx responds to different glucose concentrations to form varying concentrations of gluconic acid. This results in a decrease in the internal pH in the hydrogels. In order to increase the swelling degree of SF hydrogel in response to the decrease of pH, it is necessary to develop hydrogels with low pH. However, when the isoelectric point of SF is 4.1 and the pH is above 4.1, the swelling of SF decreases with the decrease of pH. In order to obtain a hydrogel with pH-responsiveness, the isoelectric point needs to be raised above 7, providing the SF a positive charge. Therefore, in this paper, ε-PLL is used to chemically modify SF to effect a change from negative to positive charge. In addition, it was also found that SF/ε-PLL hydrogels formed by simple blending do not have pH-responsiveness due to the lack of sufficient electrostatic repulsion. The cationic silk fibroin (CSF) hydrogel was obtained through covalent crosslinking and gelation. In order to improve the swelling degree, an SF hydrogel with random coil structure was obtained using enzymatic covalent crosslinking with HRP and H_2_O_2_ as crosslinking agents to prevent the crystallization of SF. SEM shows that this hydrogel has a highly porous architecture, providing a swelling degree in excess of 340%. Compared with self-formed hydrogels, the enzyme-triggered hydrogels have more di-tyrosine cross-linking points [56], less β-sheet cross-linking points and reveal random coil conformations. Therefore, the enzyme-triggered hydrogels exhibit excellent mechanical properties and swelling properties.

The pH-responsive mechanism of the hydrogel is shown in Figure 1 (left). The backbone chains of SF are grafted with side chains with -NH_2_ groups, and the entire side is charge-free. In the presence of H^+^, -NH_2_ can be transformed into -NH_3_^+^. In solutions with higher acidity, i.e., lower pH, due to an increase of H^+^, the electrostatic interaction expands chain spacing, thus absorbing water and showing higher swelling. When the pH increases (more basic), the swelling response decreases and pH-responsive swelling only occurs at pH < 7. In these low swelling pH hydrogels, a glucose sensitive element GOx, can specifically catalyze the formation of gluconic acid from β-D-glucose, resulting in the increase of H^+^ in the local microenvironment of the gel (Figure 1 right). At high glucose concentration, more H^+^ is formed owing to the enzymatic reaction. Due to the dense mesh entanglement, a reduced pH microenvironment is formed inside the hydrogel. The concentration of H^+^ increases, while the pH of the external environment remained unchanged. Figure 9d strongly illustrates this point. Due to the grafting of ε-PLL on the side chains of CSF, with the decrease of pH, the cationic -NH_3_^+^ group in ε-PLL increased, the electrostatic repulsion force of the gel increased, the intercellular gap between the porous crosslinked network structure increased, and the swelling degree increased. This completes the response of the gel swelling degree to the glucose concentration signal as shown in Figure 9e.

## 4. Materials and Methods

### 4.1. Experimental Materials

Materials were sourced as noted. Fresh silkworm cocoon shells: Suzhou Xiancan silk Biotechnology Co., Ltd. (Suzhou, China); Na_2_CO_3_, Na_2_HCO_3_, concentrated hydrochloric acid, sodium hydroxide, hydrogen peroxide, citric acid, disodium hydrogen phosphate, sodium dodecyl sulfate; glucose: Sinopharm Chemical Reagent Co., Ltd. (Shanghai, China); ε-PLL (molecular weight of 5000 Da), 1-ethyl-3 (3-dimethylaminopropyl) carbodiimide (EDC): Suzhou Smerford Biotechnology Co., Ltd. (Suzhou, China); LiBr: Tiancheng Chemical Co., Ltd. (Shandong, China); N-hydroxysuccinimide(NHS): Shanghai Diber Biotechnology Co., Ltd. (Shanghai, China); MES: Shanghai Alading Biotechnology Co., Ltd. (Shanghai, China); dialysis bag (molecular weight cut-off 8–10 kDa): Shanghai Yipu Biotechnology Co., Ltd. (Shanghai, China); horseradish peroxidase (activity ≥ 300 U/mg); glucose oxidase (from black liquor): Aladdin Industrial Corporation, (Shanghai, China). 

### 4.2. Preparation of Silk Fibroin Solution

The cocoons were placed in 4000 mL of 0.3 wt.% Na_2_CO_3_, 0.1 wt.% NaHCO_3_ solution and boiled at 100 °C (Figure 10). The process was repeated three times and the cocoons were scrubbed to remove the sericin completely. After washing and drying at 60 °C, the dried SF fibers were dissolved in 9.3 M LiBr solution for 1 h. The dissolved SF solution was dialyzed with deionized water at 4 °C for 3 days. After being taken out, the solution was filtered using absorbent cotton and centrifuged. The obtained SF solution was stored at 4 °C.

### 4.3. Preparation of CSF

The cationic modification of silk fibroin was carried out in accordance with existing protocols with minor modifications [58,59]. Six 150 mL beakers were taken to prepare silkworm silk fibroin solution at a concentration of 30 mg/mL. The beakers were stabilized to 1–2 °C in an ice bath, and a small amount of 0.1M morphorin-ethyl sulfonic acid (MES) was added to stabilize the pH value of the solution to about 5.5. 5 wt.% EDC (relative to silk fibroin mass) was added to all solutions. 2.5 wt.% of NHS (relative to silk fibroin mass) was added to all solutions and stirred for 30 min. ε-PLL solutions (30 mg/mL) at 0, 1, 2, 5, 10 and 15 wt.% (relative to the weight of silk fibroin) were slowly added to the above solutions, and the pH was stabilized to about 7 by using 0.1 M MES. Following reaction for 4 h in an ice bath, the solutions were placed overnight in a refrigerator at 4 °C. Dialysis (8–10 kDa) was conducted using deionized water at 4 °C for 3 days, with 12 changes of water. After dialysis, absorbent cotton was used for filtration. The solutions were centrifuged at 3000 r/min for 5 min and the supernatant solution was taken to obtain grafted cationic silk fibroin (CSF).

### 4.4. Preparation of Hydrogels 

The preparation followed commonly used protocols for enzymatic crosslinked hydrogels [56,60]. CSF solutions at 1, 2, 5, 10 and 15 wt.% were prepared as described above. HRP and H_2_O_2_ were added. In the mixed system, the solid con-tent of CSF was 3 wt.%, the HRP concentration was 10 U/mL, the H_2_O_2_ concentration was 1 mM, and the total volume was 4 mL. The solution was mixed evenly and placed at room temperature, until it naturally formed CSF hydrogel (Figure 10c). SF/ε-PLL hydrogel in the control group was obtained by blending SF solution without treatment, adding ε-PLL and then adding the same amount of HRP/H_2_O_2_ (Figure 10b). SF hydrogel in the control group was obtained by adding SF solution without treatment and spontaneously forming a gel at room temperature without HRP/H_2_O_2_ (Figure 10a).

From the stimuli responsive hydrogels, CSF solution obtained from 10 wt.% was used as above, except for the addition of GOx. When HRP, H_2_O_2_ and GOx were added into the mixture, the solid content of SF was 3 wt.%, the con-centration of HRP was 10 U/mL, the concentration of H_2_O_2_ was 1 mM, the concentration of GOx was 0, 2 and 4 mg/mL, and the total system volume was 4 mL. The solution was mixed evenly and placed at room temperature, until it naturally formed a hydrogel (Figure 10f). SF/ε-PLL/GOx hydrogel in the control group was obtained by blending the SF solution without treatment, adding ε-PLL and then adding the same amount of HRP/H_2_O_2_; the concentration of GOx was 4 mg/mL, and the total system volume was 4 mL (Figure 10e). SF/GOx in the control group was obtained by blending SF solution with GOx at room temperature. The dosage of GOx was 4 mg/mL (Figure 10d). The nomenclature used to indicate the different hydrogels studied is noted in Figure 10.

### 4.5. Characterization of CSF

#### 4.5.1. Zeta Potential

Measurements were conducted using the standard protocol [36]. 1 mL SF solution (10 mg/mL) was added to the measuring cell of a Malvern Zetasizer Nano ZS90 (Malvern instruments, Malvern, UK) at 25 °C (*n* = 3).

#### 4.5.2. Isoelectric Point

SF solution before and after treatment was pH adjusted using 0.1 M NaOH and 0.1 M HCl in the range of 5–9. The solution of 8 pH points was selected and placed in the measuring pool of the Malvern potential meter (*n* = 3). The method from 4.5.1 was used.

#### 4.5.3. Gel Electrophoresis Test

Using an existing method [61], the sample of CSF solution was properly distributed, added to SDS-PAGE buffer at the ratio of 5 to 1, boiled for 5 min, mixed and placed into the gel. The protein standard was added first, followed by a positive electrophoresis buffer to the outer tank and electrophoresis conducted at 40 V for 2 h 10 min. The gel was peeled, stained with Coomassie brilliant blue for 1 h and decolorized until the background was clear, ~2 h. The molecular weight distribution of the sample was compared to a standard protein marker.

### 4.6. Structural Measurement of CSF

#### 4.6.1. Fourier Transform Infrared Absorption Spectroscopy

The SF and CSF solution and A, B, C hydrogel samples were rapidly frozen using liquid nitrogen and dried in a freeze dryer (Virtis Genesis25-LE, Virtis Genesis25-LE Freeze Dryer). They were then shredded to a powdery form (sieving removal of diameter > 80). Samples were prepared by the KBr compression method. The infrared absorption spectrum was obtained by scanning the absorbance in the range of 400–4000 cm^−1^ in a Fourier transform infrared spectrometer (Nicolet 5700 FT-IR, Thermo Nicolet Corporation, Waltham, MA, USA)).

#### 4.6.2. Crystal Structure of CSF

The SF and CSF solution and A, B and C hydrogel samples were selected. Samples were obtained through the same treatment in Section 4.6.1 and pressed into the sample frame. The wide angle XRD pattern of the samples was obtained and analyzed by an automatic intelligent X ray diffractometer (X’PERT PRO MPD, Bruker Corporation, Berlin, Germany). Using Cuk α ray, 35 Ma stable tube current, 40 kV stable tube voltage, 8°/min scanning speed as the measurement environment, the diffraction intensity curve between 2θ = 5°−45° was scanned.

### 4.7. Morphology of Hydrogels

A scanning electron microscope (SEM) (S-4800, Hitachi, Tokyo, Japan) was used to observe the internal morphology of the hydrogel. The CSF hydrogel C and the pure SF hydrogel A (3% solid) was used as a control group. The cells were quickly frozen in liquid nitrogen and then placed in a vacuum dryer for 48 h. The dry hydrogel was sectioned, and the surface was sputtered with gold (~90 s).

### 4.8. Mechanical Properties of The Hydrogels

#### 4.8.1. Compressive Mechanical Properties

Different hydrogels were prepared and pure SF hydrogel A was used as a control. Standard samples with diameter of 10 mm and height of 8 mm were used [62]. Samples were tested for compression using a texture analyzer (TMS-PRO TM3030; Food Technology Corporation Sterling, Sterling, VA, USA) in the dry state (lift arm trial speed, 10 mm min^−1^; trigger force, 0.03 N; compression, 80%). The average of 12 parallel samples in each group was taken.

#### 4.8.2. Hydrogel Compression Resilience Test

Different hydrogels were studied, with pure SF hydrogel A as a control. Standard samples were prepared as noted in Section 4.8.1, and the compression resilience was tested. The two compression deformations were 80% of the original height of the sample. Pure SF group hydrogel samples are easy to rupture, and the compression deformation was set to 40% of the original height. The experimental initiation force, compression velocity and compression deformation are constant. The average of 12 parallel samples in each group was taken.

### 4.9. pH Sensitive Swelling Properties

In order to explore the pH swelling properties of the CSF hydrogel (C), pure SF hydrogel (A) and SF/ε-PLL hydrogel (B) were used as control group. The hydrogels (A, B and C) were placed in a constant temperature and humidity environment at 24 h to allow dry film formation. We refer to these as xerogels (xerogels mentioned subsequently are similarly prepared). The xerogel weight was M0, and the xerogel of the original weight were placed in a 50 mL centrifuge tube. Citric acid and disodium hydrogen phosphate were mixed in different proportions to form 0.2 M buffer solution with different pH values. The xerogel was added to the buffer solution (40 mL) at different pH values (pH = 4, 5, 6, 7 and 9) according to the bath ratio of 1:100 in a constant temperature water bath at 37 °C. After 6 h, the sample was removed, the surface moisture of the film was dried with filter paper and the wet weight *M*_1_ was recorded. The swelling rate is calculated by the swelling rate formula:
$$\mathrm{SR}=\frac{M_{1}-M_{0}}{M_{0}}\times 100\%$$
where *M*_1_ is the wet weight of gel, and *M*_0_ is the dry weight of gel. 

### 4.10. Performance of Glucose-Responsive Hydrogel

#### 4.10.1. Response of Internal pH of Hydrogel to External Glucose Concentration

CSF/GOx hydrogel F containing 10 wt.% ε-PLL was wrapped on a pH gauge to measure the dynamic change of pH. The hydrogels were incubated in different concentrations of glucose (0, 100, 200 and 400 mg/dL). The pH of the hydrogel was monitored (Professional Benchtop pH meter, BP3001) for a specified time (30, 60, 90, 120, 150, 180 and 240 min).

#### 4.10.2. Glucose Responsive Properties 


(1)The response of xerogel internal pH to external glucose concentration.


The hydrogel F was wrapped on a pH gauge and placed in a constant temperature and humidity environment for 24 h. A xerogel was formed on the pH probe. The xerogels were incubated at 37 °C in different concentrations of glucose (0, 100, 200 and 400 mg/dL). The pH of gels at different times were measured (Benchtop pH meter, BP3001), and the change of the pH of the external glucose solution at the same time point was monitored by another pH meter. Detection was performed at 10, 30, 60, 90, 120, 150, 180, 210, 240 and 270 min.


(2)The response of xerogel swelling degree to glucose concentration


The GOx hydrogels (D) and CSF/ε-PLL/GOx hydrogels (E) were added as the control group. Referring to the above and the glucose-responsive hydrogel swelling test [6,63], the hydrogels D, E and F were placed in a constant temperature and humidity environment, and dried to form xerogels (24 h). The results showed that the CSF/GOx hydrogel could be used as a drying agent. The original weight of xerogel was M0, and the xerogel weighing the original weight were put into a 50 mL centrifuge tube. The xerogels were incubated at 37 °C at 40 mL in different concentrations of glucose (0, 100, 200 and 400 mg/dL). After that, they were put into a constant temperature water bath of 6 h at 37 °C. After removal, the water was dried on the surface of the hydrogel with filter paper and the wet weight M1 was recorded. Three parallel samples were tested with each glucose concentration. The swelling rate was calculated as above.

## 5. Conclusions

In this work, we successfully grafted ε-PLL with silk fibroin and cross-linked it with HRP/H_2_O_2_ to form a pH-responsive cationic hydrogel. On this basis, a multifunctional pH and glucose-responsive hydrogel was obtained by incorporating GOX into the hydrogel during the preparation process. The isoelectric point of SF was changed from 4.1 to 8.6. The swelling degree of the hydrogel increases with the decrease of pH, and increased with an increase of glucose concentration. Further, this hydrogel has good mechanical properties. Considering the good biocompatibility of silk fibroin and the pH-responsive swelling of this chemically modified silk fibroin hydrogel reported in this paper, this hydrogel will be applied in drug carrier and intelligent drug transportation. At the same time, it is expected to determine the intelligent drug delivery potential, biocompatibility and antimicrobial properties of the material in in vivo and in vitro insulin release tests, cytotoxicity tests and antimicrobial tests.

## Figures and Tables

**Figure 1 ijms-22-07107-f001:**
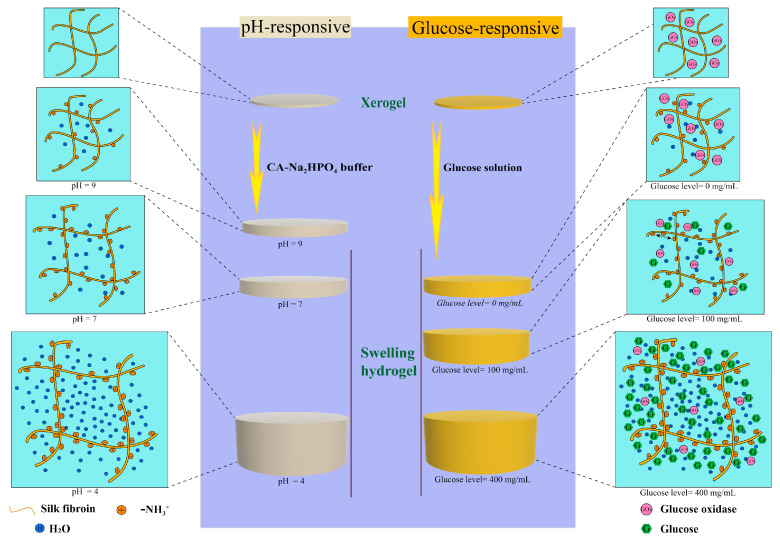
Proposed working principle of the pH-responsive swelling hydrogel (**left**) and glucose-responsive swelling hydrogel (**right**).

**Figure 2 ijms-22-07107-f002:**
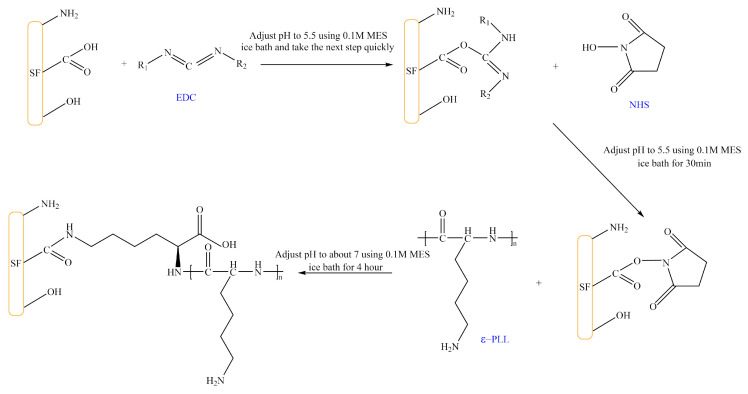
Reaction of SF grafted with ε-PLL. (The solid content of SF and CSF solution is 3 wt.%, EDC concentration is 20 mg/mL, NHS concentration is 15 mg/mL, MES concentration is 0.1 M, ε-PLL concentration is 30 mg/mL).

**Figure 3 ijms-22-07107-f003:**
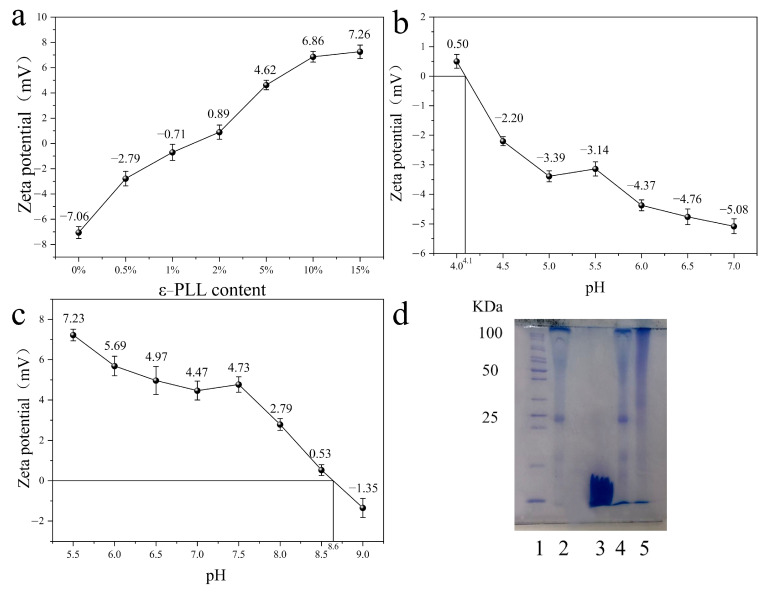
Identification of CSF: (**a**) the zeta potential of SF modified with different ε-PLL content; (**b**,**c**) Zeta potential of pure SF (**b**), CSF (**c**) under different pH conditions; (**d**) gel electrophoresis of SF before and after cationic polymerization.

**Figure 4 ijms-22-07107-f004:**
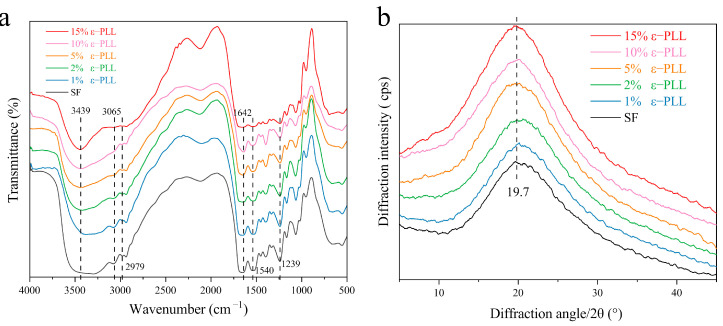
Structure of CSF: (**a**) Infrared spectrum; (**b**) X-ray diffraction curve.

**Figure 5 ijms-22-07107-f005:**
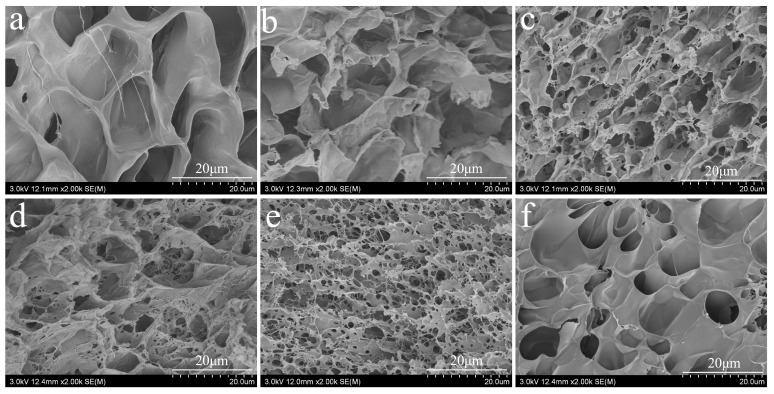
The cross-sectional morphology of CSF hydrogel: (**a**) pure SF, (**b**–**f**) the amount of ε-PLL added 1 wt.%, 2 wt.%, 5 wt.%, 10 wt.% and 15 wt.%. The solid content of SF and CSF solution is 3 wt.%, HRP concentration is 10 U/mL and H_2_O_2_ concentration is 1 mM).

**Figure 6 ijms-22-07107-f006:**
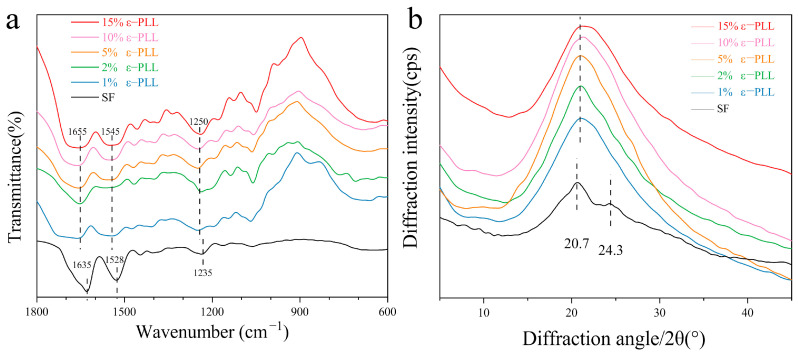
The structure of SF hydrogels: (**a**) FTIR absorption spectra; (**b**) X ray diffraction curves. (The solid content of SF and CSF solution is 3 wt.%, HRP concentration is 10 U/mL, H_2_O_2_ concentration is 1 mM).

**Figure 7 ijms-22-07107-f007:**
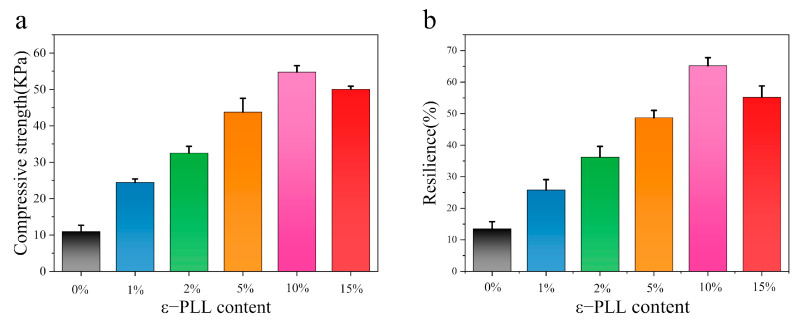
The mechanical properties of the hydrogel: (**a**) compressive strength; (**b**) compressive resilience. (The solid content of SF and CSF solution is 3 wt.%, HRP concentration is 10 U/mL, H_2_O_2_ concentration is 1 mM. *n* = 12 samples).

**Figure 8 ijms-22-07107-f008:**
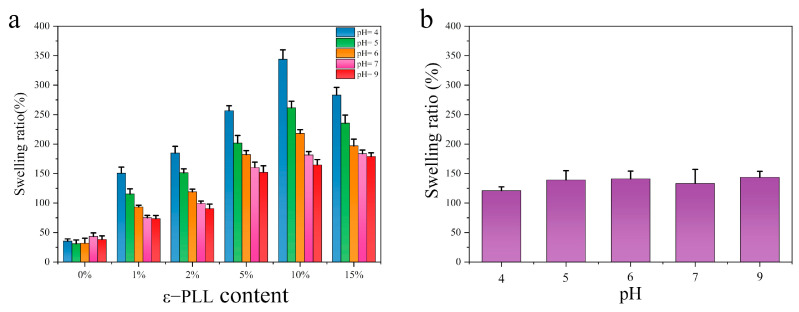
Swelling ratio of hydrogel at different pH: (**a**) CSF hydrogel (The solid content of SF and CSF solution is 3 wt.%, HRP concentration is 10 U/mL, H_2_O_2_ concentration is 1 mM); (**b**) SF/ε-PLL hydrogel (ε-PLL content: 10 wt.%). *n* = 3.

**Figure 9 ijms-22-07107-f009:**
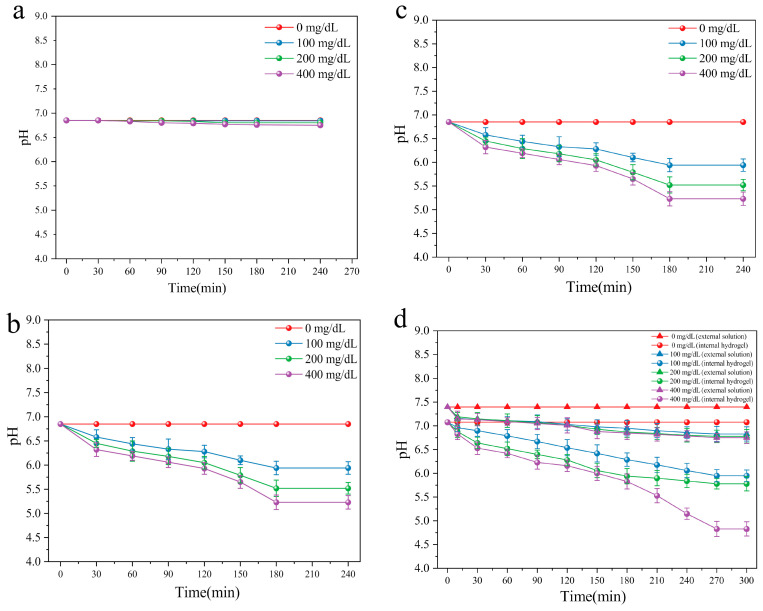
Glucose responsive properties of hydrogel: (**a**–**c**) the effect of glucose oxidase addition on the pH value of hydrogels (**a**): 0 mg/mL, (**b**): 2 mg/mL, (**c**): 4 mg/mL; (**d**) the internal and external glucose signals of xerogel were converted to pH signals; (**e**) the swelling of hydrogels added with GOx at different glucose concentrations. (The solid content of SF and CSF solution is 3 wt.%, HRP concentration is 10 U/mL, H_2_O_2_ concentration is 1 mM), *n* = 3.

**Figure 10 ijms-22-07107-f010:**
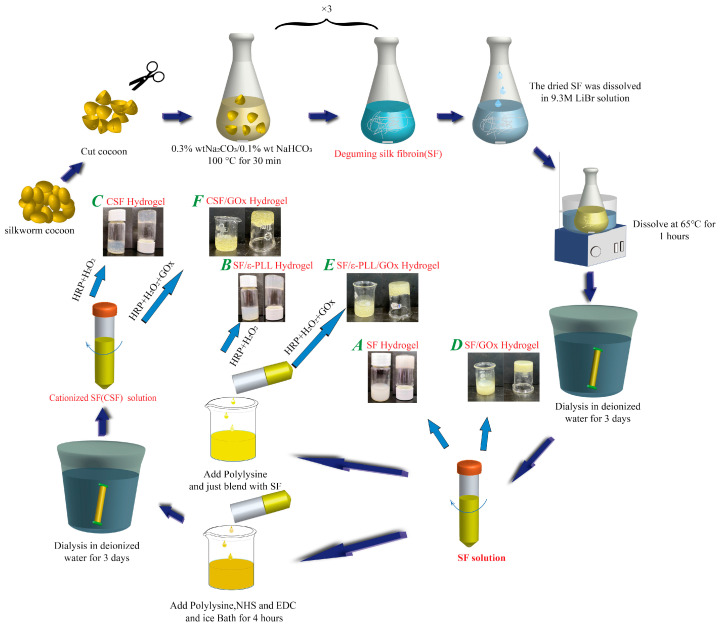
Preparation process of different hydrogels (**A**) SF hydrogel, (**B**) SF/ε-PLL hydrogel, (**C**) CSF hydrogel, (**D**) SF/GOx hydrogel, (**E**) SF/ε-PLL/GOx hydrogel, (**F**) CSF/GOx hydrogel. (The solid content of SF and CSF solution is 3 wt.%, HRP concentration is 10 U/mL, H_2_O_2_ concentration is 1 mM, GOx concentration is 0, 2 or 4 mg/mL).

## Data Availability

Not applicable.

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
