# Peer review of "Synthesis of pH and Glucose Responsive Silk Fibroin Hydrogels"

_ijms, 2021, doi:10.3390/ijms22137107_

Round 1

Reviewer 1 Report

Authors report on the synthesis and characterization of pH responsive cationic silk fibroin (SF) obtained by the chemical modification of SF with ε-Poly-(L-lysine) (ε-PLL) and enzymatic crosslinking using horseradish peroxidase and hydrogen peroxide. The topic is interesting as the hydrogel can be used for a controlled release of insulin. However, the results are not clear and the manuscript is very badly written.

Major revisions are suggested as reported below.

Preparation method is not clear, the authors never mention what is the crosslinking effect of HRP catalyzed step.

Results section is very badly written, results must be better organized and described, the section must be completely rewritten.

Some details:

  • Zeta potential must be repeated to identify the y axis in panel b of figure 2.
  • Caption of figure 2 must be rewritten, the panel do not show the PI but the zeta potential as a function of pH.
  • Line 109, sentence must be checked, do both IR peaks decrease?
  • Line 118, it is not absorption but diffraction peak.
  • Line 121, it is not “quality” but “percentage” maybe
  • Lines 118-123, the authors do not put any care in writing many repetitions are present and the whole paragraph sounds very elementary.
  • Lines 144-150, the authors do not put any care in writing many repetitions are present and the whole paragraph sounds very elementary.
  • Authors write that there is no swelling in the SF/PLL blended hydrogels, this looks unexpected , don’t you have repulsions in the blended system as well? Authors should explain the different behavior between the blended and the crosslinked polymer hydrogel.

In the preparation section, what is the meaning of “cationic CSF” in the section 4.4?

Author Response

Comments:

Authors report on the synthesis and characterization of pH responsive cationic silk fibroin (SF) obtained by the chemical modification of SF with ε-Poly-(L-lysine) (ε-PLL) and enzymatic crosslinking using horseradish peroxidase and hydrogen peroxide. The topic is interesting as the hydrogel can be used for a controlled release of insulin. However, the results are not clear and the manuscript is very badly written.

Response: We are grateful to the esteemed reviewer for providing the comments and suggestions. In response, we have edited the manuscript extensively to improve the presentation and to clarify the results and novelty of this work.

Major revisions are suggested as reported below.

Preparation method is not clear, the authors never mention what is the crosslinking effect of HRP catalyzed step.

Response: We apologize for this oversight. The text has been extensively revised to address the concern. Please see section 4.4 Preparation of hydrogels, page 11 of the revised manuscript, lines 298-315, and page 6, lines 1–20.

Text insertion: 4.4 Preparation of hydrogels

The preparation followed commonly used protocols for enzymatic crosslinked hydrogels.[59, 63] CSF solutions at 1, 2, 5, 10 and 15 wt.% were prepared as described above. HRP and H2O2 were added. In the mixed system, the solid con-tent of CSF was 3 wt.%, the HRP concentration was 10 U/mL, the H2O2 concentration was 1 mM, and the total volume was 4 mL. The solution was mixed evenly and placed at room temperature, until it naturally formed CSF hydrogel (Figure 10c). SF/ε-PLL hydrogel in the control group was obtained by blending SF solution without treatment, adding ε-PLL, and then adding the same amount of HRP/ H2O2 (Figure 10b). SF hydrogel in the control group was obtained by adding SF solution without treatment, and spontaneously forming a gel at room temperature without HRP/ H2O2 (Figure 10a).

To from the stimuli responsive hydrogels, CSF solution obtained from 10 wt.% was used as above, except for the addition of GOx. When HRP, H2O2 and GOx were added into the mixture, the solid content of SF was 3 wt.%, the con-centration of HRP was 10 U/ml, the concentration of H2O2 was 1 mM, the concentration of GOx was 0, 2 and 4 mg/ml, and the total system volume was 4 ml. The solution was mixed evenly and placed at room temperature, until it naturally formed a hydrogel (Figure 10f). SF/ε-PLL/GOx hydrogel in the control group was obtained by blending SF solution without treatment, adding ε-PLL, and then adding the same amount of HRP/ H2O2, the concentration of GOx was 4 mg/ml, and the total system volume was 4 ml (Figure 10e). SF/GOx in the control group was obtained by blending SF solution with GOx at room temperature. The dosage of GOx was 4 mg/ml (Figure 10d). The nomenclature used to indicate the different hydrogels studied is noted in Figure 10.

Regarding the crosslinking effect of the HRP catalyzed step, we have focused on the mechanical properties and reswelling performance. We have compared the differences between enzymatic cross-linked and pure filament hydrogels in both Sections 2.4 and 2.5. In order to emphasize the crosslinking effect of HRP catalyzed step, a sentence has been included in the first paragraph of section 3. Discussion, page 9 of the revised manuscript, lines 254–259.

Text insertion: Compared with self-formed hydrogels, the enzyme-triggered hydrogels have more di-tyrosine cross-linking points,[59] less β-sheet cross-linking points, and reveal random coil conformations. Therefore, the enzyme-triggered hydrogels exhibit excellent mechanical properties and swelling properties.

Results section is very badly written, results must be better organized and described, the section must be completely rewritten.

Response: Thank you for your careful review. We regret the presentation and inconvenience caused in reading. The manuscript has been thoroughly revised and rewritten, and we hope it can meet the journal’s standard.

Text insertion: In this work, we successfully grafted ε-PLL with silk fibroin and cross-linked it with HRP/H2O2 to form a pH-responsive cationic hydrogel. On this basis, a multifunctional pH and glucose-responsive hydrogel was obtained by adding GOX into the hydrogel during the preparation process. The isoelectric point of SF was changed from 4.1 to 8.6. The swelling degree of the hydrogel increased with the decrease of pH, and increased with an increase of glucose concentration. This hydrogel has good mechanical properties. Considering the good biocompatibility of silk fibroin and the pH responsive swelling of this chemically modified silk fibroin hydrogel reported in this paper, this hydrogel will be applied in drug carrier and intelligent drug transportation. At the same time, it is expected to determine the intelligent drug delivery potential, biocompatibility and antimicrobial properties of the material in vivo and in vitro insulin release tests, cytotoxicity tests and antimicrobial tests.

Some details:

Zeta potential must be repeated to identify the y axis in panel b of figure 2.

Response: Thank you for the reminder. The Y axis title of Figure 3b was inadvertently concealed during merging of figures, and has been corrected.

Caption of figure 2 must be rewritten, the panel do not show the PI but the zeta potential as a function of pH.

Response: Thank you for the reminder. The title of Figure 2 has been edited to make it more consistent with the content.

Line 109, sentence must be checked, do both IR peaks decrease?

Response: We are grateful for the suggestion and have changed the description in page 3, line 106.

Text insertion: As seen in Figure 4a, the characteristic peaks of pure SF at 2979 cm-1, 3065 cm-1 belong to the -COOH group (3300-2500 cm-1).[36, 49] After the SF was grafted with different content of ε-PLL, the characteristic peak of -COOH group (3065 cm-1) gradually decreased. The characteristic peak of amino group (3500-3300 cm-1) [50] gradually becomes sharper with an increase in mass fraction of ε-PLL.

Line 118, it is not absorption but diffraction peak.

Response: We apologize for this and has been rectified page 4, line 123,124 and page 6, line 159.

Line 121, it is not “quality” but “percentage” maybe

Response: This has been modified in page 4, line 125

Lines 118-123, the authors do not put any care in writing many repetitions are present and the whole paragraph sounds very elementary.

Response: Thank you for the careful review. The manuscript has been thoroughly revised and rewritten, so we hope it can meet the journal’s standard.

Text insertion: It can be seen from Figure 4b that pure SF is mainly a random coil structure, with a large wide diffraction peak at 20.7° and a small diffraction peak at 9.1°, indicating that SF formed a small amount of Silk II crystal structure. Compared with the pure SF diffraction curve, with the increase of ε-PLL percentage, the diffraction peak of SF remains unchanged, and a larger diffraction peak is formed at 20.7°, indicating that CSF is mainly a kind of irregular coil structure[55, 56]. The results show that ε-PLL modification has no significant effect on the crystal structure of SF.

Lines 144-150, the authors do not put any care in writing many repetitions are present and the whole paragraph sounds very elementary.

Response: Thank you for the careful review. The manuscript has been thoroughly revised and rewritten, so we hope it can meet the journal’s standard.

Text insertion: In order to study the structure change of SF/ε-PLL hydrogels, the infrared absorption spectra of these materials were studied. For example, Figure 6a shows that pure SF hydrogels have distinct absorption peaks at 1635 cm-1, 1528 cm-1, 1250 cm-1, corresponding to the absorption peaks of amide I, amide II, and amide III, which are typical characteristic peaks of β-sheet structure [55, 58]. The absorption peaks of SF/ε-PLL hydrogel at the range of 1655 cm-1, 1545 cm-1, 1235 cm-1 were obvious as the absorption peaks of amide I, amide II and III, which were mainly caused by C=O tensile vibration, N-H bending vibration and C-N tensile vibration. The results showed that the cerebrospinal fluid hydrogel prepared in this paper was mainly of random coil structure [25].

Authors write that there is no swelling in the SF/PLL blended hydrogels, this looks unexpected , don’t you have repulsions in the blended system as well? Authors should explain the different behavior between the blended and the crosslinked polymer hydrogel.

Response: Thank you for this observation. The discussion regarding this question is presented following: there should be a repulsive force. We believe that this is because the electrostatic repulsion between silk protein and poly-lysine in the blend system is far less than that between the chain repulsion of cationic silk fibroin modified by chemical grafting, so that it is difficult to show the pH-responsive swelling phenomenon in the macroscopic view. For example, Xu et al. (Polymers 2019, 11(12), 1980) have reported that chitosan-grafted silk fibroin are pH-sensitive, while no reports have been made of the pH-sensitivity of blend silk fibroin and chitosan hydrogels. At the same time, we added our relevant discussion in Section 3.Discussion, page 8, line 247-252.

Text insertion: In order to obtain a hydrogel with low pH-responsiveness, the isoelectric point of the silk fibroin protein needs to be raised above 7, giving the silk fibroin a positive charge. Therefore, in this paper, ε-PLL is used to chemically modify silk fibroin to make silk fibroin change from negative charge to positive charge, so as to prepare silk fibroin molecules with positive charge. In addition, it was also found that SF/ε-PLL hydrogels with simple blends did not have pH-responsiveness due to the lack of sufficient electrostatic repulsion.

In the preparation section, what is the meaning of “cationic CSF” in the section 4.4?

Response: We apologize for the issues in the original manuscript. In our definition, CSF stands for Cationic Silk Fibroin. The "cationic" here is superfluous and we have modified it.

Reviewer 2 Report

The authors present an interesting and logical manuscript on new silk fibroin hydrogels. In their usual form, silk fibroin hydrogels contain anionic carboxylic acid groups. By switching the anionic carboxylate groups for cationic lysine residues by chemical modification, the hydrogel becomes more biologically relevant, providing a useful pH response (and a related GOx enzymatic response owing to D-glucose oxidation to D-gluconic acid), with the hydrogels swelling at lowered pH. It follows that the hydrogels containing glucose oxidase are also responsive (increased swelling) to glucose, and therefore the system has potential for glucose-triggered insulin release/delivery.

I would recommend this paper for publication in IJMS, after the following points have been addressed:

Minor:

  1. figure 1: NHS is drawn incorrectly - a carbon-carbon double bond is shown instead of a carbon-oxygen double bond.
  2. figure 10: This is a well-presented figure. However, twice there is written 'deiomized' instead of 'deionized'.
  3. The authors have done a good job in defining the abbreviations the first time they are mentioned in the text. However, I feel that an additional complete abbreviations section at the end of the manuscript would benefit the reader.

Major:

1. The introduction of the paper implies that the ultimate aim of this study is to develop a glucose-responsive insulin delivery system for sufferers of diabetes. From this point of view, the paper would benefit from some experiments further related to the biological applications. For example, cytotoxicity experiments, or glucose-triggered insulin delivery/release studies in vitro or an in vivo model. I feel that at the very least, there should be some extra explanation added to the discussion / conclusions regarding the future direction of the work, and that this work is still more of a proof of concept at this stage, with more work needed to prove the drug delivery applications and biocompatibility.

Author Response

Comments:

The authors present an interesting and logical manuscript on new silk fibroin hydrogels. In their usual form, silk fibroin hydrogels contain anionic carboxylic acid groups. By switching the anionic carboxylate groups for cationic lysine residues by chemical modification, the hydrogel becomes more biologically relevant, providing a useful pH response (and a related GOx enzymatic response owing to D-glucose oxidation to D-gluconic acid), with the hydrogels swelling at lowered pH. It follows that the hydrogels containing glucose oxidase are also responsive (increased swelling) to glucose, and therefore the system has potential for glucose-triggered insulin release/delivery.

Response: We appreciate the reviewer’s positive evaluation of our work for taking the time to review this manuscript. Please find the revisions in the re-submitted files.

I would recommend this paper for publication in IJMS, after the following points have been addressed:

Minor:

  1. figure 1: NHS is drawn incorrectly - a carbon-carbon double bond is shown instead of a carbon-oxygen double bond.

Response: We are grateful to the reviewer for pointing this out and have fixed the issue.

  1. figure 10: This is a well-presented figure. However, twice there is written 'deiomized' instead of 'deionized'

Response: We are grateful to the reviewer for pointing out the error and have modified the figure accordingly.

The authors have done a good job in defining the abbreviations the first time they are mentioned in the text. However, I feel that an additional complete abbreviations section at the end of the manuscript would benefit the reader.

Response: Thank you for your suggestion. As noted, the suggested content in the form of a table has been added to the end of the manuscript.(page 17)

Text insertion:

List of abbreviation

SF

Silk Fibroin

CSF

Cationic Silk Fibroin

GOx

glucose oxidase

ε-PLL

ε-Poly-(L-lysine)

NHS

N-hydroxysuccinimide

EDC

1-ethyl-3 (3-dimethylaminopropyl) carbodiimide

MES

morpholine ethylsulfonic acid

XRD

X-ray diffraction

FTIR

Fourier transform infrared spectroscopy

SEM

Scanning electron microscope

pI

isoelectric point

  1. The introduction of the paper implies that the ultimate aim of this study is to develop a glucose-responsive insulin delivery system for sufferers of diabetes. From this point of view, the paper would benefit from some experiments further related to the biological applications. For example, cytotoxicity experiments, or glucose-triggered insulin delivery/release studies in vitro or an in vivo model. I feel that at the very least, there should be some extra explanation added to the discussion / conclusions regarding the future direction of the work, and that this work is still more of a proof of concept at this stage, with more work needed to prove the drug delivery applications and biocompatibility.

Response: We appreciate the reviewer’s positive evaluation of our work and agree with the comments regarding the limitations of our study. Our current work has led to the development of a novel material that could have biologically relevant applications in the future. We believe that in the future, insulin delivery studies in vivo and in vitro and related cytotoxicity tests can be successfully carried out as you mentioned. We added an outlook section to the section 5.Conclusion for further work, in the hope that our future work will demonstrate the potential of drug delivery applications and biocompatibility of this material.(page 12, line 409).

Text insertion: Considering the excellent biocompatibility of silk fibroin and the pH responsive swelling of this chemically modified silk fibroin hydrogel reported in this paper, this hydrogel has the potential to be used as a drug carrier and for intelligent drug transportation. Future work is expected to determine the biocompatibility, cytotoxicity and antimicrobial properties of the material for controlled in vivo and in vitro insulin release.

Reviewer 3 Report

Synthesis of pH and glucose responsive silk fibroin hydrogels

Manuscript ID: ijms-1264206

In the present manuscript Tao et al., used Poly lysine (PLL) to render pH responsive behavior to form silk fibroin (SF) hydrogels. The investigators have modified SF with PLL and studied their physicochemical properties of the hydrogels. In addition, the authors have explored the responsiveness of the SF hydrogels to glucose concentration via Glucose Oxidase (GOx) addition. This is a good study and should be considered for publication in IJMS after major revision.

My comments are:

  • The authors have mentioned Isoelectric point of SF as 4.2 (line 73) and as 4.1 (line 94). Please be consistent.
  • The authors have used only zeta potential study to confirm PLL modification to SF. Probably, the authors can provide the NMR of the modified SF and compare it with regenerated SF.
  • Line 9 (Abstract): The authors wrote: we present a novel composite viz. cationic SF (CSF) obtained by the chemical crosslinking modification of SF with ε-Poly-(L-lysine) (ε-PLL)

-The authors used ‘composite’ word for PLL modified SF. I believe it should be ‘conjugate’ instead of composite since it is chemically modified. Also, in the same line the authors wrote cationic SF was obtained by chemical crosslinking modification of SF with PLL. However, there is no crosslinking here, but only chemical modification (carbodiimide coupling; Figure 1) of PLL with SF. The authors should only write chemical modification.

  • The figure captions need to be descriptive and more detailed. e.g. in Figure 1, the authors used EDC/NHS to activate the -COOH. In the figure caption, the authors should use all reaction conditions (all reagents, buffer concentration, pH, time, temperature) for more clarity to the readers. Same comment goes for all figures. The authors should note sample size, error calculations and all in the figure caption where appropriate. Also in the chemical modification of PLL onto SF, there are some other reports by Kaplan lab where they reported PLL modification onto Silk fibroin. They should be cited. (Biomacromolecules2020, 21, 7, 2829–2843, Biomacromolecules 2010, 11, 12, 3406–3412)
  • Preparation of CSF (Page 9): the preparation protocol should be more clear, in the figure 1, it seems the authors have added EDC first and then NHS, however, in the method, it seems the authors have added NHS before EDC. Please make it consistent. Also, please mention how many dialysis water changes are made, how was dialysis performed (Mol. Wt of Dialysis tube).
  • Preparation of CSF hydrogels: I got little confused with the protocol here. Probably, the authors need to rephrase here and clearly mention the control gels, the sample gels, conditions and concentrations of all reactants and enzymes.
  • Figure 10 caption is very short and it is difficult to understand the message of the authors. Probably, the authors can be more descriptive here.
  • Similarly, for gel electrophoresis, the authors need to mention details about the gel they used, the protein standard and conditions. I expect the 30 minute boiled silk will have molecular weight above 100 kDa (many reports from Kaplan lab), however, it seems the authors used a protein standard of upto 100 kDa. This is also apparent from the gel image (Figure 2D), since it is visible there is still some material which did not move at all (probably due to their high molecular weight). It would be great if authors use a different protein standard and do SDS Page (probably use methods in the paper:
    Sci., 2020,8, 4176-4185 if possible)
  • About the discussion of FTIR: The authors wrote, ‘In order to study the effect of grafting on the aggregation of SF, the infrared absorption spectrum and XRD before and after grafting were characterized. As seen in Figure 3a, the characteristic peaks of pure SF at 2979 cm-1 , 3065 cm-1 107 belong to the -COOH group (3300-2500 cm-1). After the SF was grafted with different content of ε-PLL, the characteristic peak of -COOH group (3300-2500 cm-1) gradually decreased, while that at 3439 cm-1 decreased.’

 - There is apparent decrease in 3439 cm-1 peak, however, it becomes sharper with increase in grafting density. The authors needs to revise this and also should refer articles regarding FTIR peaks of -COOH and -NH2 to back their claims.

  • The authors found chemical grafting of PLL to silk fibroin doesn’t change the aggregation/conformation of silk fibroin which is an interesting finding and consistent with recent report (Advanced Biology, 2021, 2100388 DOI: 1002/adbi.202100388) where they found chemistry do not significantly affect the structural integrity of silk polymer (by calculating the percentage beta-sheet content). The authors can add a beta-sheet quantification to the study or refer the paper to support their finding.
  • For FTIR studies of CSF hydrogels, the authors find less beta-sheets (Figure 5) and more random coil as PLL content increase. It would be great if authors can quantify beta-sheets and other secondary structures.
  • What is the motivation of adding glucose and glucose oxidase to already prepared hydrogels (HRP/H2O2). The text is little confusing, as I am not sure if glucose/GOx was added to already prepared CSF gel or in solution.
  • Figure 8 legends need to be bigger and clear.
  • While the pH and Glucose responsiveness of the hydrogel can be achieved by adding glucose and GOx, what was the need for HRP/H2O2.
  • Statement in conclusion: CSF hydrogels were obtained by covalently crosslinking ε-PLL on the SF. I am not sure if it is correct. I guess, the authors form CSF hydrogels by exposing the chemically modified SF-PLL construct to HRP and H2O2. I do not see any crosslinking between PLL and SF.
  • Also in the conclusions, where the authors stated different possible application of these hydrogels in drug delivery, intelligent drug transport etc, should be supported by few references.

Author Response

Comments:

In the present manuscript Tao et al., used Poly lysine (PLL) to render pH responsive behavior to form silk fibroin (SF) hydrogels. The investigators have modified SF with PLL and studied their physicochemical properties of the hydrogels. In addition, the authors have explored the responsiveness of the SF hydrogels to glucose concentration via Glucose Oxidase (GOx) addition. This is a good study and should be considered for publication in IJMS after major revision.

Response: We thank the reviewer for the summary and really appreciate the efforts in reviewing the manuscript. The manuscript has been revised in accordance with the comments of the reviewer and the point-by-point responses are detailed below.

My comments are:

The authors have mentioned Isoelectric point of SF as 4.2 (line 73) and as 4.1 (line 94). Please be consistent.

Response: We apologize for this and have corrected all clerical errors (marked in red).

The authors have used only zeta potential study to confirm PLL modification to SF. Probably, the authors can provide the NMR of the modified SF and compare it with regenerated SF.

Response: We believe that the presented characterization including the FTIR, XRD, gel electrophoresis, and zeta-potential measurements confirm the modification, together with the observations of the study itself. Through the comparison of the properties of SF/ε-PLL blends with the following hydrogels, we believe that PLL successfully modified silk fibroin protein. Owing to time and other considerations, we may not be able to supplement this part of the experiment and believe that it may be beyond the scope of this present work.

Line 9 (Abstract): The authors wrote: we present a novel composite viz. cationic SF (CSF) obtained by the chemical crosslinking modification of SF with ε-Poly-(L-lysine) (ε-PLL)

-The authors used ‘composite’ word for PLL modified SF. I believe it should be ‘conjugate’ instead of composite since it is chemically modified. Also, in the same line the authors wrote cationic SF was obtained by chemical crosslinking modification of SF with PLL. However, there is no crosslinking here, but only chemical modification (carbodiimide coupling; Figure 1) of PLL with SF. The authors should only write chemical modification.

Response: We thank the reviewer for the reminder. The sentence containing "composite" has been verified to make sure it is used correctly.

Text insertion: In order to imbue stimuli responsive behavior in silk fibroin, we propose a new conjugated material, namely cationic SF (CSF) obtained by chemical modification of silk fibroin with ε-Poly-(L-lysine) (ε-PLL).

The figure captions need to be descriptive and more detailed. e.g. in Figure 1, the authors used EDC/NHS to activate the -COOH. In the figure caption, the authors should use all reaction conditions (all reagents, buffer concentration, pH, time, temperature) for more clarity to the readers. Same comment goes for all figures. The authors should note sample size, error calculations and all in the figure caption where appropriate. Also in the chemical modification of PLL onto SF, there are some other reports by Kaplan lab where they reported PLL modification onto Silk fibroin. They should be cited. (Biomacromolecules2020, 21, 7, 2829–2843, Biomacromolecules 2010, 11, 12, 3406–3412)

Response: We are grateful for the careful work and thoughtful suggestions that have helped improve this paper substantially. Figure 1 and its title have been modified to ensure that the reaction conditions (all reagents, buffer concentration, pH, time, temperature) are shown in the title,(page 3, line 83-84)

Text insertion:

Figure 2. Reaction of SF grafted with ε-PLL. (The solid content of SF and CSF solution is 3 wt.%, EDC concentration is 20 mg/mL, NHS concentration is 15 mg/mL, MES concentration is 0.1 M, ε-PLL concentration is 30 mg/mL)

Other relevant figure titles have been supplemented appropriately. The relevant results of Kaplan laboratory are referred to in the chemical modification experiment. The relevant changes have been highlighted in red in the original text.(page 10, line 289-290)

Text insertion: The cationic modification of silk fibroin was carried out in accordance with existing protocols with minor modifications.[61, 62]

Preparation of CSF (Page 9): the preparation protocol should be more clear, in the figure 1, it seems the authors have added EDC first and then NHS, however, in the method, it seems the authors have added NHS before EDC. Please make it consistent. Also, please mention how many dialysis water changes are made, how was dialysis performed (Mol. Wt of Dialysis tube).

Response: We are extremely grateful to the reviewer for pointing out this concern. This information has been presented in Section 4.3 Preparation of CSF, highlighted in red.(Page 10)

Preparation of CSF hydrogels: I got little confused with the protocol here. Probably, the authors need to rephrase here and clearly mention the control gels, the sample gels, conditions and concentrations of all reactants and enzymes.

Response: We appreciate the reviewer’s suggestion. According to the reviewer’s comment, more details have been provided and in the rewritten Section 4.4 Preparation of hydrogel, highlighted in red.(page 11, line 309-325)

Text insertion:

4.4 Preparation of hydrogels

The preparation followed commonly used protocols for enzymatic crosslinked hydrogels.[59, 63] CSF solutions at 1, 2, 5, 10 and 15 wt.% were prepared as described above. HRP and H2O2 were added. In the mixed system, the solid con-tent of CSF was 3 wt.%, the HRP concentration was 10 U/mL, the H2O2 concentration was 1 mM, and the total volume was 4 mL. The solution was mixed evenly and placed at room temperature, until it naturally formed CSF hydrogel (Figure 10c). SF/ε-PLL hydrogel in the control group was obtained by blending SF solution without treatment, adding ε-PLL, and then adding the same amount of HRP/ H2O2 (Figure 10b). SF hydrogel in the control group was obtained by adding SF solution without treatment, and spontaneously forming a gel at room temperature without HRP/ H2O2 (Figure 10a).

To from the stimuli responsive hydrogels, CSF solution obtained from 10 wt.% was used as above, except for the addition of GOx. When HRP, H2O2 and GOx were added into the mixture, the solid content of SF was 3 wt.%, the con-centration of HRP was 10 U/ml, the concentration of H2O2 was 1 mM, the concentration of GOx was 0, 2 and 4 mg/ml, and the total system volume was 4 ml. The solution was mixed evenly and placed at room temperature, until it natural-ly formed a hydrogel (Figure 10f). SF/ε-PLL/GOx hydrogel in the control group was obtained by blending SF solution without treatment, adding ε-PLL, and then adding the same amount of HRP/ H2O2, the concentration of GOx was 4 mg/ml, and the total system volume was 4 ml (Figure 10e). SF/GOx in the control group was obtained by blending SF solution with GOx at room temperature. The dosage of GOx was 4 mg/ml (Figure 10d). The nomenclature used to indi-cate the different hydrogels studied is noted in Figure 10.

Figure 10 caption is very short and it is difficult to understand the message of the authors. Probably, the authors can be more descriptive here.

Response: Thank you very much for your valuable comments. Our original intention was to show our sample preparation process through this figure, so as to avoid plain text difficult for readers to understand directly. The specific information is located in the text above the figure. According to the suggestion, we have added some content under the figure title to make it contain more information and highlighted it in red.(page 12, line 327-329)

Text insertion:

Figure 10. Preparation process of different hydrogels (A) SF hydrogel , (B) SF/ε-PLL hydrogel, (C) CSF hydrogel, (D) SF/GOx hydrogel, (E) SF/ε-PLL/GOx hydrogel, (F) CSF/GOx hydrogel. (The solid content of SF and CSF solution is 3 wt.%, HRP concentration is 10 U/mL, H2O2 concentration is 1 mM, GOx concentration is 0, 2 or 4 mg/mL)

Similarly, for gel electrophoresis, the authors need to mention details about the gel they used, the protein standard and conditions. I expect the 30 minute boiled silk will have molecular weight above 100 kDa (many reports from Kaplan lab), however, it seems the authors used a protein standard of upto 100 kDa. This is also apparent from the gel image (Figure 2D), since it is visible there is still some material which did not move at all (probably due to their high molecular weight). It would be great if authors use a different protein standard and do SDS Page (probably use methods in the paper: Sci., 2020,8, 4176-4185 if possible)

Response: We are grateful for the suggestion and questions about gel electrophoresis experiments, and for mentioning that SDS PAGE can be done using different protein standards, which is definitely correct. We will definitely try to use the method in the literature mentioned to conduct gel electrophoresis test in the future, which can indeed better display the overall molecular weight distribution. However, our maximum protein standard of 100 kDa can also prove that we have successfully carried out chemical modification. For the low molecular weight region, the changes are more clearly visible and help distinguish the differences between silk fibroin protein blends and chemical modification treatments.

About the discussion of FTIR: The authors wrote, ‘In order to study the effect of grafting on the aggregation of SF, the infrared absorption spectrum and XRD before and after grafting were characterized. As seen in Figure 3a, the characteristic peaks of pure SF at 2979 cm-1 , 3065 cm-1 107 belong to the -COOH group (3300-2500 cm-1). After the SF was grafted with different content of ε-PLL, the characteristic peak of -COOH group (3300-2500 cm-1) gradually decreased, while that at 3439 cm-1 decreased.’

- There is apparent decrease in 3439 cm-1 peak, however, it becomes sharper with increase in grafting density. The authors needs to revise this and also should refer articles regarding FTIR peaks of -COOH and -NH2 to back their claims.

Response: We deeply appreciate the reviewer’s suggestion. According to the reviewer’s comment, We have added several more detailed papers on the FTIR peaks of -COOH and -NH2 to support our claim have highlighted them in red in the original text. At the same time, the description of the peak change of 3439 cm-1 was also modified.(page 4, line109-112)

Text insertion: As seen in Figure 4a, the characteristic peaks of pure SF at 2979 cm-1, 3065 cm-1 belong to the -COOH group (3300-2500 cm-1)[36, 49]. After the SF was grafted with different content of ε-PLL, the characteristic peak of -COOH group (3065 cm-1) gradually decreased. The characteristic peak of amino group (3500-3300 cm-1)[50] gradually becomes sharper with an increase of the mass fraction of ε-PLL.

The authors found chemical grafting of PLL to silk fibroin doesn’t change the aggregation/conformation of silk fibroin which is an interesting finding and consistent with recent report (Advanced Biology, 2021, 2100388 DOI: 1002/adbi.202100388) where they found chemistry do not significantly affect the structural integrity of silk polymer (by calculating the percentage beta-sheet content). The authors can add a beta-sheet quantification to the study or refer the paper to support their finding.

Response: Thank you for the careful review and suggestion. This important piece of research found that chemical grafting to silk fibroin doesn’t change the aggregation/conformation of silk fibroin. Our experimental results undoubtedly strongly prove this conclusion and we have added this reference to the section 2.1.

Text insertion: Our results are in excellent agreement with this phenomenon which was previously reported by Sahoo et al.[54]

For FTIR studies of CSF hydrogels, the authors find less beta-sheets (Figure 5) and more random coil as PLL content increase. It would be great if authors can quantify beta-sheets and other secondary structures.

Response: We are grateful to the reviewer for the careful work and thoughtful suggestions that have helped improve this paper substantially. At the same time, we believe that the results characterized by the aggregation structure of hydrogels in this paper indicate that the enzymatic cross-linked hydrogels mainly exhibit the random coiled structure conformation, which is completely different from the self-formed hydrogels which exhibit the β-sheet structure conformation. This structure provides the basis for its pH and glucose responsiveness, and also provides excellent mechanical properties of hydrogels. We regret that we failed to pay attention to the change of β-sheet with the increase of ε-PLL content. This is because in the FTIR results, we focused on the overall results presented, and the specific β-sheet change results play a limited role here. At the same time, in our previous simple quantitative analysis, we found that with the increase of ε-PLL content, the change of β-sheet was not obvious, and the difference between enzymatic cross-linked hydrogels and self-formed hydrogels had a great influence. This phenomenon can be determined only by observing the wavelength position of the maximum absorption peak without quantitative analysis

What is the motivation of adding glucose and glucose oxidase to already prepared hydrogels (HRP/H2O2). The text is little confusing, as I am not sure if glucose/GOx was added to already prepared CSF gel or in solution.

Response: We appreciate your questions and apologize for the confusion but believe we have corrected these questions - we have rewritten the Section 4.4 Preparation of hydrogels.(page 11, line 309-325)

Text insertion: 4.4 Preparation of hydrogels

The preparation followed commonly used protocols for enzymatic crosslinked hydrogels.[59, 63] CSF solutions at 1, 2, 5, 10 and 15 wt.% were prepared as described above. HRP and H2O2 were added. In the mixed system, the solid con-tent of CSF was 3 wt.%, the HRP concentration was 10 U/mL, the H2O2 concentration was 1 mM, and the total volume was 4 mL. The solution was mixed evenly and placed at room temperature, until it naturally formed CSF hydrogel (Figure 10c). SF/ε-PLL hydrogel in the control group was obtained by blending SF solution without treatment, adding ε-PLL, and then adding the same amount of HRP/ H2O2 (Figure 10b). SF hydrogel in the control group was obtained by adding SF solution without treatment, and spontaneously forming a gel at room temperature without HRP/ H2O2 (Figure 10a).

To from the stimuli responsive hydrogels, CSF solution obtained from 10 wt.% was used as above, except for the addition of GOx. When HRP, H2O2 and GOx were added into the mixture, the solid content of SF was 3 wt.%, the con-centration of HRP was 10 U/ml, the concentration of H2O2 was 1 mM, the concentration of GOx was 0, 2 and 4 mg/ml, and the total system volume was 4 ml. The solution was mixed evenly and placed at room temperature, until it natural-ly formed a hydrogel (Figure 10f). SF/ε-PLL/GOx hydrogel in the control group was obtained by blending SF solution without treatment, adding ε-PLL, and then adding the same amount of HRP/ H2O2, the concentration of GOx was 4 mg/ml, and the total system volume was 4 ml (Figure 10e). SF/GOx in the control group was obtained by blending SF solution with GOx at room temperature. The dosage of GOx was 4 mg/ml (Figure 10d). The nomenclature used to indi-cate the different hydrogels studied is noted in Figure 10.

In the preparation of hydrogel containing GOX, glucose oxidase was added in the solution state (without gel formation) and mixed evenly in the solution system to form hydrogel containing GOX.

Figure 8 legends need to be bigger and clear.

Response: Thank you for underlining this deficiency. The higher definition versions of the images have been updated. We believe that we can provide the editor with clearer pictures to meet the requirements of the journal in the later official publication

While the pH and Glucose responsiveness of the hydrogel can be achieved by adding glucose and GOx, what was the need for HRP/H2O2.

Statement in conclusion: CSF hydrogels were obtained by covalently crosslinking ε-PLL on the SF. I am not sure if it is correct. I guess, the authors form CSF hydrogels by exposing the chemically modified SF-PLL construct to HRP and H2O2. I do not see any crosslinking between PLL and SF.

Response: Our deepest gratitude goes to you for your careful work and thoughtful suggestions that have helped improve this paper substantially. We have modified our expression in the original text and marked it in red.(page 14, line 411-412)

Text insertion: In this work, we successfully grafted ε-PLL with silk fibroin (SF) and cross-linked it with HRP/H2O2 to form a pH-responsive cationic hydrogel (CSF).

It is the covalent cross-linking of tyrosine on CSF caused by HRP/H2O2 to form hydrogel rather than the covalent cross-linking between ε-PLL.

Also in the conclusions, where the authors stated different possible application of these hydrogels in drug delivery, intelligent drug transport etc, should be supported by few references.

Response: Thank you for your summary. We really appreciate your efforts in reviewing our manuscript. We have revised the manuscript accordingly. In the section 5. Conclusions,(page 14, line 415-419) we look forward to some possible applications of the new material,

Text insertion: Considering the good biocompatibility of silk fibroin and the pH responsive swelling of this chemically modified silk fibroin hydrogel reported in this paper, this hydrogel will be applied in drug carrier and intelligent drug transportation. At the same time, it is expected to determine the intelligent drug delivery potential, biocompatibility and antimicrobial properties of the material in in vivo and in vitro insulin release tests, cytotoxicity tests and antimicrobial tests.

In the last paragraph of the section 1.Introduction, (page 2, line 68-70) we add some references, hoping to fully explain the application potential of the new material prepared and synthesized by us.

Text insertion: The rapid diffusion mechanism driven by high swelling provides the possibility for the intelligent transport of drugs, and the possibility for the closed-loop detection and treatment of diabetes in vivo.[47,48]

Reviewer 4 Report

I have reviewed a manuscript entitled “Synthesis of pH and glucose responsive silk fibroin hydrogels”. It is a very interesting work, aiming to prepare ε-Poly-(L-lysine)-grafted SF hydrogels with stimuli-responsive properties. I think it is suitable for publication after addressing some minor comments:

Comment 1: I would suggest moving figure 9 to after introduction.

Comment 2: please use consistent labeling for all the figures. For example, in figure 4, the labels are red while in other figures are white.

Comment 3: while you have claimed the antibacterial activity of ε-Poly-(L-lysine), there is no experiment showing the antibacterial activity of developed SF hydrogel.

Comment 4: what are the potential application of this developed hydrogel?

Author Response

Comments:

I have reviewed a manuscript entitled “Synthesis of pH and glucose responsive silk fibroin hydrogels”. It is a very interesting work, aiming to prepare ε-Poly-(L-lysine)-grafted SF hydrogels with stimuli-responsive properties. I think it is suitable for publication after addressing some minor

Response: Thank you for your letter and the reviewers’ comments concerning our manuscript. Those comments are valuable and very helpful. We have read through comments carefully and have made corrections. Based on the instructions provided in your letter, we uploaded the file of the revised manuscript. Revisions in the text are shown using red highlight for additions. The responses to the reviewer's comments are marked in blue and presented following.

Comment 1: I would suggest moving figure 9 to after introduction.

Response: Thank you very much for your suggestion on our paper. The figure placed in this way helps to describe the observed pH responsiveness and glucose responsiveness swelling principle. We have added Figure 9 to the introduction, and hope to keep figure 9 in the discussion section to make the content of the discussion section easier to understand.

Comment 2: please use consistent labeling for all the figures. For example, in figure 4, the labels are red while in other figures are white.

Response: We are sorry that the theme of figure 4 is not consistent. Since the SEM image itself is black and white, we re-used the color matching (white) in the revised manuscript to make it clearer.

Comment 3: while you have claimed the antibacterial activity of ε-Poly-(L-lysine), there is no experiment showing the antibacterial activity of developed SF hydrogel.

Response: Thank you for your careful review. We regret that we did not carry out relevant experimental studies to confirm this statement. However, based on prior reported results, including the following references (Adv. Funct. Mater. 2017, 27, 1604894; Biomacromolecules 2018 19 (2), 279-287), we believe that ε-PLL has certain antibacterial potential. Meanwhile, we added some references to the section 1.Introduction and highlighted them in red.(page 2, line 61-62) We will fully examine the antimicrobial potential for future drug loads or other biological applications, and add this prospect to the section 5.Conclusion. We hope to conduct in vivo and in vitro drug delivery studies, related cytotoxicity tests and antimicrobial properties tests while conducting application studies.(page 14, line 417-419)

Text insertion: At the same time, it is expected to determine the intelligent drug delivery potential, biocompatibility and antimicrobial properties of the material in in vivo and in vitro insulin release tests, cytotoxicity tests and antimicrobial tests.

Comment 4: what are the potential application of this developed hydrogel?

Response: We expect that the hydrogel can be used as a drug carrier material, such as insulin loading. It has a higher swelling rate at high glucose and releases insulin quickly, a lower swelling rate at low glucose or no insulin release at all. Of course, this is only our experimental conjecture, is still in the verification. As an intelligent and responsive material, it can also be used in other biological fields such as diagnosis of disease and routine monitoring.

Round 2

Reviewer 2 Report

The authors have satisfactorily addressed the queries and I believe the manuscript is now acceptable for publication in IJMS.

Reviewer 3 Report

The authors have significantly improved the manuscript post revision and revised most of the suggestions. I would recommend accepting the article post minor english typos edit